# Structural Analysis and Anti-Inflammatory Effect of a Digalactosyldiacylglycerol-Monoestolide, a Characteristic Glycolipid in Oats

**DOI:** 10.3390/nu14194153

**Published:** 2022-10-06

**Authors:** Hiroki Yamada, Junya Ito, Naoki Shimizu, Takumi Takahashi, Chikara Kato, Isabella Supardi Parida, Mirinthorn Jutanom, Katsuyuki Ishihara, Kiyotaka Nakagawa

**Affiliations:** 1Laboratory of Food Function Analysis, Graduate School of Agricultural Science, Tohoku University, Sendai 980-8572, Japan; 2Department of Cell Biology, Division of Host Defense Mechanism, Tokai University School of Medicine, Isehara 259-1193, Japan; 3R&D Center, Calbee, Inc., Utsunomiya 321-3231, Japan

**Keywords:** DGDG-estolides, DGDG, FAHFA, oats, *Avena sativa*, anti-inflammatory effect

## Abstract

Digalactosyldiacylglycerol- (DGDG-) monoestolide is a characteristic glycolipid in oats. DGDG-monoestolides possess a unique structure whereby a fatty acid of DGDG is replaced by a fatty acid ester of hydroxy fatty acid (FAHFA). While the physiological effects of DGDG and FAHFA have been reported previously, the effects of DGDG-monoestolides are unknown. Hence, we isolated a major DGDG-monoestolide molecular species from oats, analyzed its structure, and evaluated its anti-inflammatory effect. Based on GC-MS, MS/MS, and NMR analyses, the isolated compound was identified as a DGDG-monoestolide that contains the linoleic acid ester of 15-hydroxy linoleic acid (LAHLA) and linoleic acid (i.e., DGDG-LAHLA). The isolated DGDG-LAHLA was evaluated for its anti-inflammatory effect on LPS-stimulated RAW264 cells. The production of nitric oxide and cytokines (IL-6, TNF-α, and IL-10) were significantly decreased by DGDG-LAHLA, suggesting the anti-inflammatory effect of DGDG-LAHLA for the first time. In addition, our data showed a pronounced uptake of DGDG-LAHLA by cells. Some compounds corresponding to the predicted DGDG-LAHLA metabolites were also detected, suggesting that both intact DGDG-LAHLA and its metabolites may contribute to the above anti-inflammatory activities. These results are expected to expand the availability of oats as a functional food.

## 1. Introduction

Oats (*Avena sativa*), a typical grain often consumed as oatmeal and granola, have attracted attention as a functional food due to their high nutritional value and content of functional compounds. For example, previous studies have demonstrated that oats-derived β-glucan reduces serum cholesterol levels [1]. Furthermore, avenanthramides, a group of phenolic compounds that are unique to oats, were shown to inhibit obesity in high-fat diet-fed mice by regulating intestinal flora and alleviating oxidative stress and inflammation [2]. Oats are also characterized by their high lipid content, which is several times higher than other grains. Nevertheless, most studies have focused on the functions of oats’ water-soluble compounds (e.g., β-glucan and avenanthramides), while reports on the effects of lipid-soluble compounds in oats remain scarce.

In the few studies concerning lipid-soluble compounds contained in oats, the presence of digalactosyldiacylglycerol (DGDG, Figure 1A) [3] and fatty acids esters of hydroxy fatty acids (FAHFA, Figure 1B) [4] has been demonstrated. Additionally, Hamberg et al. identified DGDG-monoestolide as a characteristic lipid contained in oats [5]. The identified DGDG-monoestolide consisted of a glycerolipid containing linoleic acid (LA) at the *sn*-1 position, an LA ester of 15-hydroxy linoleic acid (15-HLA) (i.e., 15-LAHLA) at the *sn*-2 position, and two galactose molecules at the *sn*-3 position (i.e., DGDG-LAHLA, Figure 1C). However, to the best of our knowledge, only one study has reported on the detailed structural analysis of DGDG-monoestolides. Furthermore, although the physiological effects (e.g., anti-inflammatory effects) of both DGDG [6,7,8]^,^ and FAHFA [4,9,10,11] have previously been reported, the effects of DGDG-monoestolides, which contain DGDG and FAHFA in their structures, have not been evaluated.

In this study, we isolated one of the major molecular species of DGDG-monoestolides from oats and analyzed its chemical structure. Based on structural analyses by GC-MS, MS, MS/MS, and NMR, the isolated compound was identified as DGDG-LAHLA (Figure 1C). We investigated the anti-inflammatory activity of DGDG-LAHLA and its cellular uptake in an effort to elucidate the mechanisms underlying its bioactivities. Our study is the first to demonstrate the anti-inflammatory and cellular uptake of DGDG-LAHLA in RAW 264 macrophage cells. This report on the physiological effects of DGDG-monoestolides reveals new insights into the functional properties of oats, which hopefully serves as a sound basis for expanding the availability of oats as a functional food.

## 2. Materials and Methods

### 2.1. Chemicals

Oats were powdered using a Wonder Crusher WC-3 (OSAKA CHEMICAL Co., Ltd., Osaka, Japan). Silica Gel 60N (spherical, neutral) was purchased from KANTO CHEMICAL Co., INC (Tokyo, Japan). Sweeley reagent was obtained from Tokyo Chemical Industry Co., Ltd. (Tokyo, Japan). Lipase from *Rhizopus oryzae* was purchased from Sigma Aldrich (St Louis, MO, USA). Lipopolysaccharide (LPS), fetal bovine serum (FBS), and minimum essential medium (MEM) non-essential amino acids solution (NEAA) were purchased from FUJIFILM Wako Pure Chemical Corporation (Osaka, Japan). WST-1 reagent was purchased from DOJINDO LABORATORIES (Kumamoto, Japan). All other reagents were of the highest grade available.

### 2.2. Isolation of a Major DGDG-Monoestolide (Compound X) from Powdered Oat

Powdered oats (100 g) were stirred in methanol (800 mL) for 2–3 h, after which chloroform (800 mL) was added and stirred overnight. The extract was filtered through filter paper (No. 5C, TOYO ROSHI KAISHA, LTD., Tokyo, Japan), and the eluate was evaporated to obtain total lipids. Total lipids were dissolved in chloroform (40 mL) and filtered through a glass filter to remove highly polar extracts. The obtained eluate was subjected to silica gel column chromatography using 240 g of silica gel and a chromatography column (64 ID × 600 Length mm, Chemglass Inc., Vineland, NJ, USA) to fractionate total lipids into lipid classes. The column was eluted with chloroform (3 L), acetone (3 L), and methanol (2 L) to obtain neutral lipid, glycolipid, and phospholipid fractions, respectively.

A portion of the glycolipid fraction was further separated by semi-preparative high-performance liquid chromatography (HPLC)-UV under Condition 1 (Table 1). Each peak detected in the UV chromatogram was collected. The collected peaks were analyzed on a micrOTOF-Q II mass spectrometer (Bruker Daltonik, Billerica, MA, USA) under the conditions described in Table 2 (Condition 1). Among the peaks that were presumed to be DGDG-monoestolides, a single major peak was defined as compound X. The remaining glycolipid fraction was fractionated under Condition 1 (Table 1) to obtain a crude fraction of compound X. Crude compound X was further purified by HPLC-UV under Condition 2 (Table 1). The obtained compound X was analyzed with HPLC-MS. The MS system consisted of a micrOTOF-Q II mass spectrometer attached to a Nexera HPLC system (Shimadzu, Kyoto, Japan). HPLC-MS conditions described in Table 1 (Condition 3) and Table 2 (Condition 1) were used.

### 2.3. Structural Analysis of Compound X

Compound X was analyzed with GC-MS, MS, MS/MS, and NMR to determine its detailed structure.

GC-MS analysis was conducted to determine the fatty acids and hydroxy fatty acids contained in compound X. A mixture of hydrochloric acid–methanol–benzene (8:1:2, *v*/*v*/*v*, 2 mL) and 0.01% BHT in methanol (5 μL) was added to 20 μg of compound X. After heating at 100 °C for 1 h, 5 mL of aqueous potassium carbonate (6%, wt%) and hexane (1 mL) were added. The mixture was centrifuged (1660× *g*, 4 °C, 5 min), and the upper layer was collected. The lower layer was reextracted with hexane (1 mL) under the same conditions. The collected upper layers were combined, evaporated, and dissolved in hexane (400 μL). A portion of this solution (100 μL) was mixed with Sweeley reagent (1 μL), and kept at room temperature for 30 min. The resultant product was analyzed with a GC-MS system consisting of a GCMS-QP2010 SE (Shimadzu) equipped with an HP-5MS UI column (length, 30 m; internal diameter, 0.25 mm; film thickness, 0.25 μm; Agilent Technologies, Santa Clara, CA, USA). The injector and detector temperatures were set at 250 °C. Helium gas was introduced as the carrier gas at a flow rate of 50 mL/min (24.8 cm/sec). The gradient profile was as follows: 150 °C for 15 min, 150–240 °C (6 °C/min linear), and 240 °C for 5 min. Compounds were identified based on a spectral library search using the NIST 17 Mass Spectral Library.

MS and MS/MS analysis was performed using a micrOTOF-Q II mass spectrometer under the conditions described in Table 2 (Conditions 1 and 2, respectively). Samples were dissolved in methanol and analyzed by direct infusion into the mass spectrometer.

NMR (^1^H, ^13^C, correlation spectroscopy (COSY), total correlation spectroscopy (TOCSY), heteronuclear single-quantum correlation spectroscopy (HSQC), heteronuclear multiple-bond correlation spectroscopy (HMBC), nuclear overhauser enhancement and exchange spectroscopy (NOESY), and rotating frame nuclear overhauser effect spectroscopy (ROESY)) spectra were recorded on a Varian 600TT (Varian Medical Systems, CA, USA) at 600 MHz using tetradeuteromethanol as a solvent.

Compound X was also treated with lipase from *Rhizopus oryzae*. Compound X was dissolved of chloroform–methanol (2:1, *v*/*v*, 20 µL) containing 0.15% (*w*/*v*) Triton X-100. The solvent was evaporated and added with 0.04 M Tris buffer (0.1 mL). Lipase from *Rhizopus oryzae* (250 units) was added, and the mixture was stirred at room temperature for 60 min. The solution was adjusted to pH 3 with 10% H_2_SO_4_. The solution was mixed with chloroform–methanol (2:1, *v*/*v*, 600 µL), vortexed for 5 min, and centrifuged (1660× *g* for 5 min at 4 °C). The lower layer was collected in a different tube. The upper layer was similarly reextracted after the addition of chloroform–methanol (10:1, *v*/*v*, 400 µL). The collected lower layers were combined, evaporated, dissolved in methanol (500 µL), and analyzed by HPLC-MS/MS. The MS system consisted of a QTRAP 6500 tandem mass spectrometer attached to an ExionLC system (SCIEX, Tokyo, Japan). HPLC-MS/MS condition described in Table 1 (Condition 5) and Table 3.

### 2.4. Cell Culture

Mouse macrophage RAW264 cells were obtained from Japan Food Research Laboratories (Tokyo, Japan). Cells were cultured in MEM (Sigma Aldrich) supplemented with 10% FBS, 1% NEAA, penicillin, and streptomycin at 37 °C with 5% CO_2_.

### 2.5. Analysis of Nitric Oxide (NO) Production and Cell Viability

RAW264 cells were seeded in 96-well plates at a density of 1.0 × 10^4^ cells/well (*n* = 6). After incubation for 24 h, 20 µL of medium containing DGDG-LAHLA and LPS was added to each well. DGDG-LAHLA used in this experiment was similar to that used in Section 2.3. (Structural analysis of compound X). Final concentrations were set to 0–5 µM of DGDG-LAHLA and 100 ng/mL of LPS. After further incubation for 24 h, 80 µL of the supernatant was transferred to another 96-well plate for analyzing the NO production. Cell viability was evaluated using the remaining cells.

To determine the amount of NO in the supernatant, Griess reagent [12] (80 µL) was added to the supernatant, and the mixture was incubated at 37 °C for 20 min. Absorbance at 532 nm was measured with a microplate reader (Infinite^®^ 200 PRO, Tecan, Männedorf, Switzerland).

Cell viability was determined using the WST-1 assay (*n* = 6). Medium (70 µL) and WST-1 reagent (10 µL) were added to the cells, after which the absorbance at 450 nm was measured. Absorbance was measured once again after incubation at 37 °C for 2–3 h.

### 2.6. Analysis of Inflammatory and Anti-Inflammatory Cytokine Production

RAW264 cells were seeded in 96-well plates at a density of 1.0 × 10^4^ cells/well for IL-6 and TNF-α, and 2.0 × 10^4^ cells/well for IL-10 (*n* = 4). After incubation for 24 h, a medium supplemented with DGDG-LAHLA and LPS was added. Final concentrations were set to 0–5 µM of DGDG-LAHLA and 100 ng/mL of LPS. After further incubation for 24 h, the medium was collected. Cytokine (i.e., IL-6, TNF-α and IL-10) production was evaluated using the ELISA MAX Deluxe Set (BioLegend, San Diego, CA, USA) following the protocols described by the manufacturer.

### 2.7. Analysis of Cellular Uptake

RAW264 cells were seeded in 6 cm dishes. After incubation for 24 h, the medium was removed, and a medium containing 5 µM of DGDG-LAHLA was added to the dish. After further incubation for 24 h, the medium was removed, and cells were washed three times with PBS, collected, and centrifuged (1630× *g*, 4 °C, 5 min). After removing the supernatant, cells were suspended in PBS (420 µL) and cell lysate was prepared with sonication. The cell lysate was then used for the cellular uptake and protein assay.

To isolate DGDG-LAHLA, cell lysate (300 µL) was first mixed with 0.9% potassium chloride solution (300 µL) and chloroform–methanol (2:1, *v*/*v*, 2.4 mL), then stirred for 5 min. The mixture was then centrifuged (1660× *g*, 4 °C, 5 min) and the lower layer was collected in a separate tube. Chloroform–methanol (10:1, *v*/*v*, 1.6 mL) was added to the upper layer, and the mixture was stirred and centrifuged under the same conditions as before. The lower layer from the second centrifugation was combined with the first one, evaporated, dissolved in chloroform (500 µL), and stored at −30 °C. In the subsequent step, the extract was subjected to a solid phase extraction (SPE) using a silica gel SPE column (100 mg, 1 mL; Phenomenex, Torrance, CA) previously conditioned with chloroform (2 mL). After loading the extract (200 µL), the column was eluted with chloroform (4 mL) and methanol (2 mL), respectively. Each fraction was evaporated and redissolved in methanol (200 µL) for HPLC-MS/MS analysis in multiple reaction monitoring (MRM) mode (HPLC conditions; Table 1, Condition 5; MS conditions Table 3). For the analysis, DGDG-LAHLA and 15-LAHLA obtained by hydrolyzing DGDG-LAHLA were used as reference standards. In addition, analyses for compounds that were expected to be generated from DGDG-LAHLA by the release of sugar and fatty acid moieties were performed with the predicted MRM pairs.

### 2.8. Protein Assay

The level of DGDG-LAHLA uptake by cells was calculated relative to the total protein content in cell lysates. The Bradford assay was used to quantify the protein content in the cell lysates. Prior to the assay, cell lysates and standard bovine serum albumin were diluted in PBS. To perform the assay in the 96-well plates, lysate or albumin standard (10 µL) was added to each well, followed by the staining solution (200 µL). The reaction mixture was then stirred for 5 s, and absorbance (595 nm) was measured with a microplate reader.

### 2.9. Statistical Analysis

Statistical analysis was performed by one-way ANOVA followed by Tukey’s test. The data are presented as mean ± standard deviation (SD). A *p* value of <0.05 was considered significant (GraphPad Prism ver.9 (San Diego, CA, USA)).

## 3. Results and Discussion

### 3.1. Isolation and Structural Analysis of a Major DGDG-Monoestolide (Compound X) Contained in Oat

DGDG-monoestolides are characteristically contained in oats and possess a unique structure whereby one of the fatty acids that constitute DGDG is replaced by FAHFA. Depending on the composition of fatty acids and FAHFA, DGDG-monoestolides can consist of various molecular species (Figure 1). Previous studies have demonstrated that DGDG and FAHFA, the constituents of DGDG-monoestolides, both possess physiological effects (e.g., anti-inflammatory effects) [4,6,8,9]. Despite that, the effects of DGDG-monoestolides have not been evaluated. Thus, in this study, we first aimed to isolate a DGDG-monoestolide from oats. Then, the isolated DGDG-monoestolide was evaluated for its anti-inflammatory effect, an effect that has been reported both for DGDG and FAHFA.

Total lipids were extracted from approximately 100 g of oats by methanol–chloroform extraction. The total lipid content was 10.9 ± 0.1 g per 100 g wet weight. Then, using silica gel column chromatography, the extracted total lipids were separated into neutral lipid, glycolipid, and phospholipid fractions. The contents of each fraction per 100 g of oats were neutral lipids, 9.1 ± 0.6 g; glycolipids, 0.6 ± 0.2 g, and phospholipids, 1.1 ± 0.1 g. This lipid composition was consistent with a previous study reporting that oats contain 10% lipids consisting mainly of neutral lipids [13]. Considering that the glycolipid content of common edible plants is 0.005–0.645 g per 100 g [14], these results demonstrate that oats contain more glycolipids than other plant species.

To further fractionate the extracted glycolipids into DGDG-monoestolide molecular species, a portion of the extracted glycolipid fraction was subjected to semi-preparative HPLC-UV (Condition 1, Table 1). Each peak detected in the UV chromatogram was collected and subjected to MS analysis. Based on the detected molecular weights of the peaks that eluted between 10–20 min, the presence of several DGDG-monoestolide molecular species was suggested. Among these peaks, we defined one of the major peaks (*m/z* 1241.8267 [M+Na]^+^) as compound X. Compound X was fractionated from the remaining glycolipid fraction by semi-preparative HPLC-UV (Table 1, condition 1). Semi-preparative HPLC-UV with different conditions was also performed (Table 1, condition 2) for further purification. HPLC-MS analysis after isolation of compound X showed a single peak, suggesting its high purity (Figure 2A,B).

Subsequently, structural analyses of compound X were performed. To determine the fatty acids contained in compound X, a portion of the isolated compound X was hydrolyzed, derivatized, and analyzed with GC-MS. Two distinct peaks were observed on the GC-MS chromatogram. By conducting a spectral library search, each peak was predicted to be the derivatized forms of LA (Rt. 16.6 min) and 15-HLA (Rt. 20.1 min) (Figure 2C). Further structural analyses were carried out by MS and MS/MS. MS analysis demonstrated a single peak at *m/z* 1241.826 [M+Na]^+^, corresponding to the molecular formula of C_69_H_116_O_19_Na (Figure 3). Hence, MS/MS analysis was performed using *m/z* 1241.826 as the precursor ion. The product ion of *m/z* 961.584 (neutral loss of 280.242 Da) corresponded to a loss of LA. Additionally, the product ion of *m/z* 683.362 (neutral loss of 558.464 Da) was presumed to correspond to the loss of 15-LAHLA, a LA ester of 15-HLA (both 15-HLA and LA were detected during the above GC-MS analysis). These results suggested that LA and 15-LAHLA were contained in compound X. Additionally, with reference to a previous study on the MS/MS analysis of DGDG [15,16], the product ion of *m/z* 1079.774 (neutral loss of 162.052 Da) was considered to be a loss of a sugar molecule. Hence, based on the above GC-MS and MS/MS analyses, compound X was suggested to be a DGDG-monoestolide molecular species containing 15-LAHLA and LA.

Finally, various 1D and 2D NMR (^1^H, ^13^C, COSY, TOCSY, HSQC, HMBC, NOESY, ROESY) analyses were performed to analyze the structure of compound X in detail (Appendix A). In the HMBC spectrum, the anomeric protons of α-galactose (4.860 ppm) and β-galactose (4.237 ppm) correlated with C-6 of β-galactose (67.800 ppm) and the *sn*-3 position of glycerol (68.781 ppm), respectively. This demonstrated that the β-galactosyl-(6→1)-α-galactose moiety was bound to the *sn*-3 position of glycerol. Additionally, COSY analysis revealed that the signal around 4.9 ppm, which overlapped with the peak of water in the sample, was an oxymethine group at C-15 of HLA. This proton correlated with a carboxyl carbon (175.295 ppm) in the HMBC spectrum, confirming that 15-LAHLA is, indeed, a component of compound X. Incidentally, treatment of compound X with a lipase from *Rhizopus oryzae* (a lipase that cleaves fatty acids at the *sn*-1 position of glycolipids [17]) also demonstrated that LA was bound to the *sn*-1 position and LAHLA was bound to the *sn*-2 position. Thus, based on the above GC-MS, MS, MS/MS, and NMR analyses, the structure of compound X was confirmed as DGDG-LAHLA with the exact structure demonstrated in Figure 1C. These results of MS and NMR analysis also showed that the purity of isolated DGDG-LAHLA reached >99%, which was dissolved in ethanol for use in a cell culture study.

As described above, various analyses were conducted to isolate and characterize DGDG-LAHLA from oats. During this process, compounds that share a similar structure with DGDG-LAHLA were also detected. For example, during the LC-MS analysis of the glycolipid fraction, *m/z* 1239, 1243, and 1245, corresponding to the molecular species of DGDG-monoestolides [18], were detected (data not shown). Additionally, *m/z* 1520 suggested the presence of DGDG-diestolides [18], a compound in which the *sn*-2 position of DGDG-monoestolides is further esterified with another HLA molecule (data not shown). Although this study was focused on the isolation and structural analysis of DGDG-LAHLA, the described methods should also be applicable to the isolation and structural analysis of the above compounds. Hence, analyses of DGDG-monoestolide molecular species and DGDG-diestolides, leading to the further elucidation of the characteristic lipid profile of oats (e.g., the ratio of DGDG-LAHLA in the DGDG-monoestolides fraction of oats), are expected in future studies.

### 3.2. Evaluation of the Anti-Inflammatory Effect of DGDG-LAHLA

As described above, the physiological effects of DGDG-monoestolides have not been evaluated to date. On the other hand, DGDG, a constituent of DGDG-monoestolides, is known to possess an anti-inflammatory effect. For example, Banskota et al. reported that the treatment of LPS-stimulated RAW264 cells with 20 μM DGDG inhibited the production of NO, an inflammatory mediator, by about 20–30% [19]. The mechanism underlying this anti-inflammatory effect was suggested to be the suppression of inducible nitric oxide synthase (iNOS) expression. Furthermore, this anti-inflammatory effect was observed regardless of the fatty acid composition of DGDG. Meanwhile, other studies have reported that FAHFA, another component of DGDG-monoestolides, also possesses an anti-inflammatory effect [4,9,10,11]. Based on these studies, we anticipated that DGDG-monoestolides, which share the structural characteristics of DGDG and FAHFA, possess a strong anti-inflammatory effect. Hence, we examined the effect of DGDG-LAHLA, a molecular species of DGDG-monoestolides, on the secretion of NO and several cytokines towards LPS-stimulated RAW264 cells.

Treatment of cells with LPS induces the production of NO. Hence, NO was measured as an indicator of inflammation [20]. LPS-stimulated RAW264 cells produced a 20-fold higher amount of NO than untreated cells, suggesting that LPS causes significant inflammation in RAW264 cells (Figure 4A). NO production was evaluated in the range of 0.5–5 μM DGDG-LAHLA, which barely had any effect on cell viability (Figure 4B). NO production in LPS-stimulated RAW264 cells was suppressed in a concentration-dependent manner when 0.5–5 μM DGDG-LAHLA was added (Figure 4A). Notably, treatment of 5 µM DGDG-LAHLA suppressed NO production by 40%, suggesting, for the first time, that DGDG-LAHLA possesses a significant anti-inflammatory effect at concentrations low as several μM. Meanwhile, DGDG and FAHFA, the constituents of DGDG-LAHLA, were effective at several tens of μM in previous studies [4,19]. Hence, the strong anti-inflammatory effect of DGDG-LAHLA may be attributed to the additive/synergistic effect of DGDG and LAHLA. In future studies, we plan to elucidate the molecular mechanisms by which DGDG-LAHLA exerts such a strong effect by evaluating the anti-inflammatory effects not only of DGDG-LAHLA but also its possible metabolites (e.g., DGDG and LAHLA).

The effects of DGDG-LAHLA on the production of IL-6, TNF-α (two major pro-inflammatory cytokines), and IL-10 (an anti-inflammatory cytokine) were evaluated. The concentrations of these cytokines secreted from RAW264 cells were significantly increased by LPS treatment, and this increase was significantly suppressed by the addition of DGDG-LAHLA (Figure 5). These results demonstrated that DGDG-LAHLA suppressed the production of both pro-inflammatory and anti-inflammatory cytokines. Such simultaneous suppression of both pro-inflammatory and anti-inflammatory cytokines has been reported for several other compounds (e.g., tribulusamide D [21] and *Mytilus coruscus* polysaccharide [22]). In these studies, the inhibition of an upstream transcription factor, NF-κB, was suggested as an underlying mechanism. We speculate that the anti-inflammatory effect of DGDG-LAHLA observed in this study is also exerted by a similar mechanism. Further studies are necessary to verify such mechanisms, and we plan to evaluate the effects of DGDG-LAHLA on NF-κB signaling activity in future studies.

### 3.3. Evaluation of the Cellular Uptake of DGDG-LAHLA

We showed, for the first time, that DGDG-LAHLA, one of the DGDG-monoestolide molecular species characteristically present in oats, has anti-inflammatory properties. Following this, we evaluated the cellular uptake of DGDG-LAHLA as it is crucial for its functional expression in cells.

To perform the uptake study, RAW264 cells were treated with 5 µM of DGDG-LAHLA, a concentration where the anti-inflammatory effect was the highest. After treatment, a distinct peak of DGDG-LAHLA was detected in the treated cells (Figure 6), indicating that DGDG-LAHLA was, indeed, taken up into the cells. Quantification using the prepared standard showed that the amount of DGDG-LAHLA in the cell lysate was 22.5 ± 0.6 pmol/mg protein. When compared to the cellular uptake of other anti-inflammatory compounds [23], this amount of uptake is considered sufficient to exert anti-inflammatory effects. The anti-inflammatory activities of DGDG-LAHLA may derive not only from the intact compound but also from its metabolites. For this reason, we attempted to predict the cellular metabolism of DGDG-LAHLA using the predicted MRM pairs of compounds that could arise from the DGDG-LAHLA metabolism (Appendix A). As a result, we propose the following pathways: (1) desorption of LA occurs at the *sn*-1 position (MRM, *m/z* 979.6 > 699.6; Compound A), (2) desorption of LA from LAHLA at the *sn*-2 position (MRM, *m/z* 979.6 > 683.6; Compound B), (3) desorption of the remaining 15-HLA at the *sn*-2 position of Compound B (MRM, *m/z* 701.4 > 539.4; Compound D). In addition, we detected both 15-HLA and 15-LAHLA, supposedly produced from the proposed pathway. In the future, accurate quantification of DGDG-LAHLA metabolites is important to identify the extent to which these metabolites contribute to the bioactivities of DGDG-LAHLA. For this, reference standards corresponding to each metabolite are required.

## 4. Conclusions

Although oats contain various functional compounds, little is known regarding the lipid-soluble compounds in oats. Of such lipid-soluble compounds, we focused on DGDG-monoestolide, a glycolipid characteristic of oats, and aimed to evaluate its physiological effects. We isolated one of the major DGDG-monoestolide molecular species in oats and determined its structure as DGDG-LAHLA by GC-MS, MS/MS, and NMR analyses. These analyses were also expected to be useful in determining the structures of similar compounds (e.g., other molecular species of DGDG-monoestolide and DGDG-diestolides). Subsequently, the anti-inflammatory effect of DGDG-LAHLA was demonstrated by evaluating NO production in LPS-stimulated RAW264 cells. To partially reveal its mechanism of action, DGDG-LAHLA was found to inhibit the production of pro-inflammatory and anti-inflammatory cytokines regulated by NF-κB. However, further studies are necessary to confirm the signaling pathways involved in the anti-inflammatory effect of DGDG-LAHLA. In addition, our results clearly show that DGDG-LAHLA is taken up by cells. It is important to determine whether the DGDG-LAHLA detected in the cell is localized in the cytoplasm or is membrane-bound. Since we did not separately analyze the DGDG-LAHLA content in the membrane and cytoplasm, we would like to investigate the cellular localization of the DGDG-LAHLA in future studies. Some compounds that were expected to be the breakdown products of DGDG-LAHLA were also detected in cells. These compounds may only be a small part of the wide array of breakdown products that are produced from the degradation of DGDG-LAHLA. Therefore, future studies should focus on expanding the analysis to include the identification and quantification of degradation products of DGDG-LAHLA. Although the issues mentioned above remain, we have shown, for the first time, that DGDG-LAHLA possesses an anti-inflammatory effect and is taken up by cells. We anticipate that these results will provide new knowledge into the functionality of the lipid-soluble compounds in oats, contributing to the use of oats as a functional food.

## Figures and Tables

**Figure 1 nutrients-14-04153-f001:**
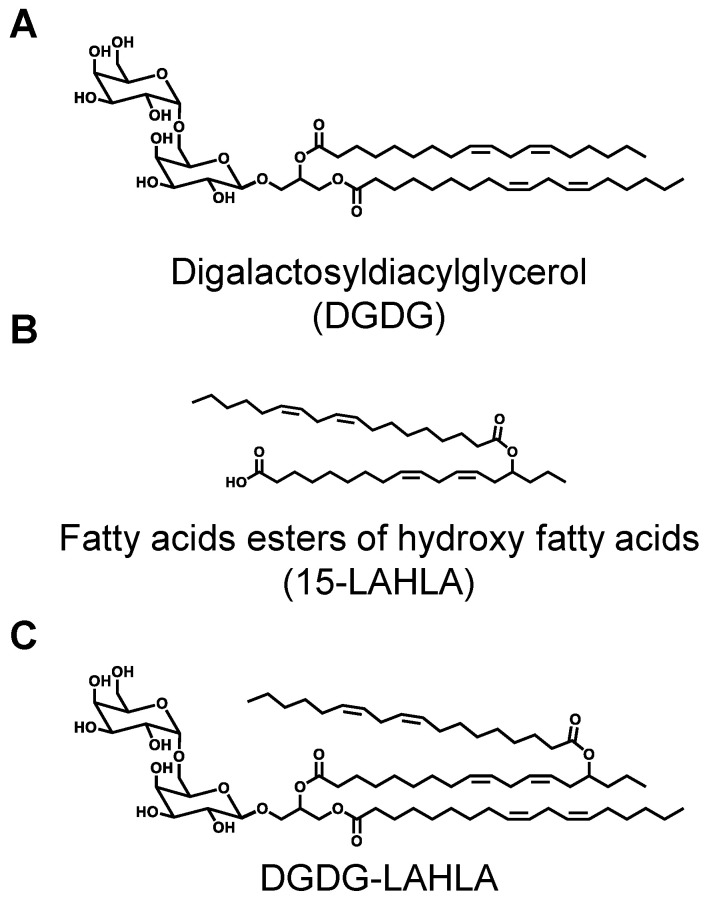
Chemical structures of (**A**) DGDG, (**B**) 15-LAHLA, and (**C**) DGDG-LAHLA. DGDG, digalactosyldiacylglycerol; LAHLA, linoleic acid ester of hydroxy linoleic acid.

**Figure 2 nutrients-14-04153-f002:**
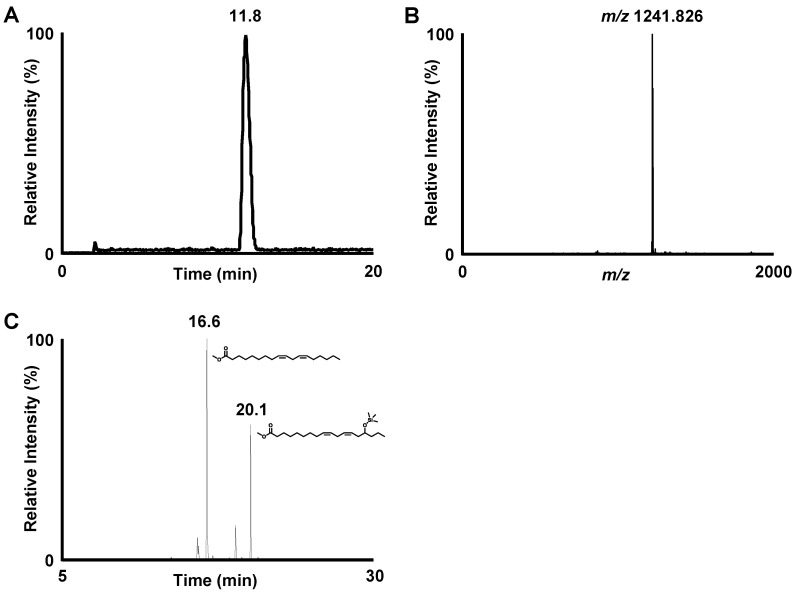
(**A**) LC-MS chromatogram of base peak chromatogram (BPC) of compound X analyzed under conditions described in Table 1 (Condition 3) and Table 2 (Condition 1), and (**B**) spectrum of the peak at 11.8 min. (**C**) GC-MS chromatogram of total ion chromatogram (TIC) of the hydrolyzed and derivatized compound X. The two peaks were identified as methyl linoleate (16.6 min) and methyl 15-hydroxy linoleate with a TMS group on the hydroxy group (20.1 min) by a spectral library search.

**Figure 3 nutrients-14-04153-f003:**
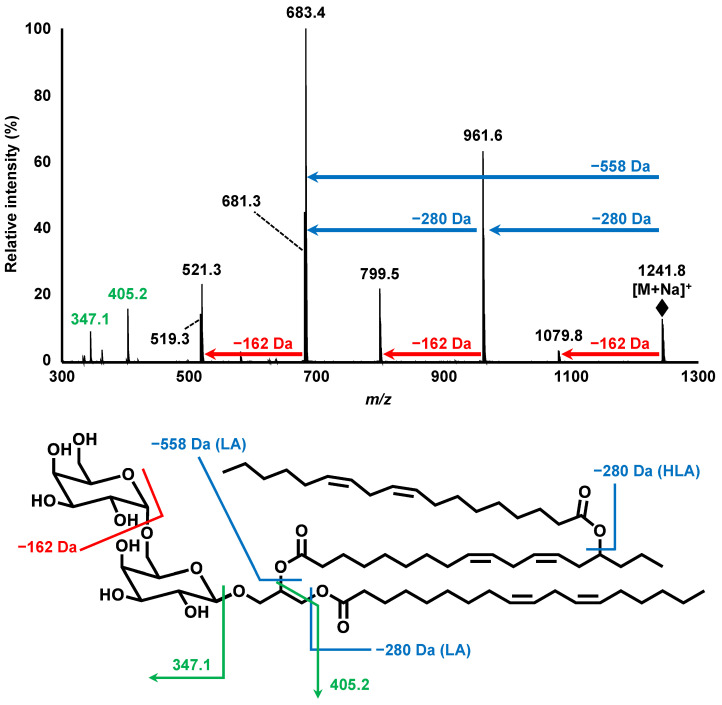
MS/MS spectra of product ion scan analysis of *m/z* 1241.826. Analytical conditions are described in Table 2 (Condition 2). LA, linoleic acid; HLA, hydroxy linoleic acid.

**Figure 4 nutrients-14-04153-f004:**
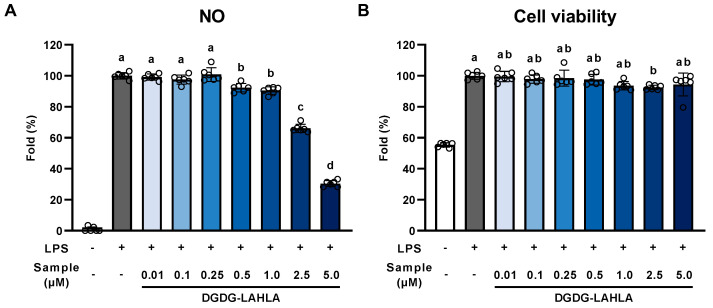
Effect of DGDG-LAHLA towards (**A**) NO production and (**B**) cell viability. RAW264 cells treated with LPS (100 ng/mL) were co-treated with DGDG-LAHLA (0.01–5 µM) and incubated for 24 h. NO in the culture supernatant was measured using Griess reagent. Cell viabilities were measured using WST-1 reagent. Data represents mean ± SD (*n* = 6). Different letters indicate significant differences (*p* < 0.05, Tukey’s test).

**Figure 5 nutrients-14-04153-f005:**
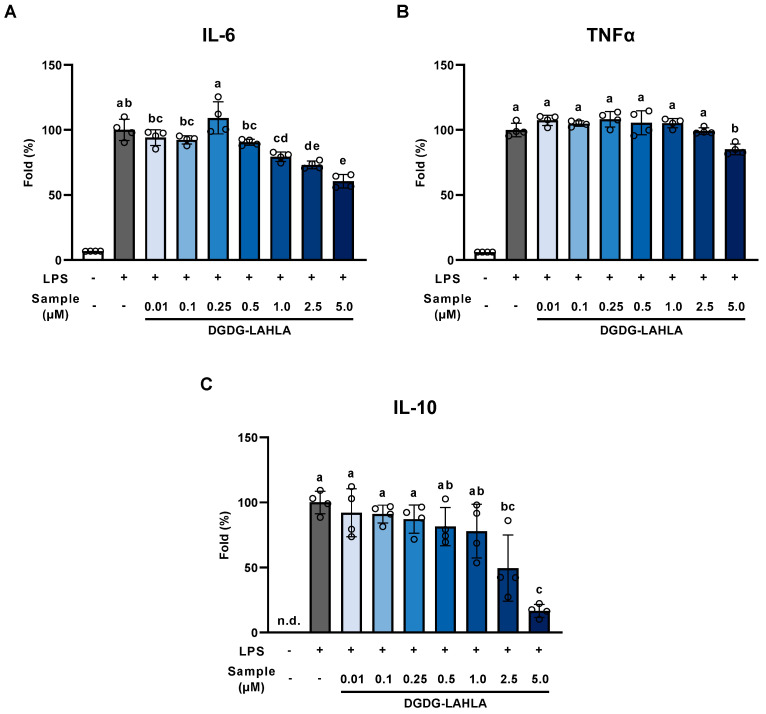
Effect of DGDG-LAHLA on the production of cytokines ((**A**) IL-6, (**B**) TNF-α, and (**C**) IL-10). RAW264 cells treated with LPS (100 ng/mL) were co-treated with DGDG-LAHLA (0.01–5 µM) and incubated for 24 h. Cytokine levels in the culture supernatant were measured by ELISA. Data represents mean ± SD (*n* = 4). Different letters indicate significant differences (*p* < 0.05, Tukey’s test). IL, interleukin; TNF, tumor necrosis factor.

**Figure 6 nutrients-14-04153-f006:**
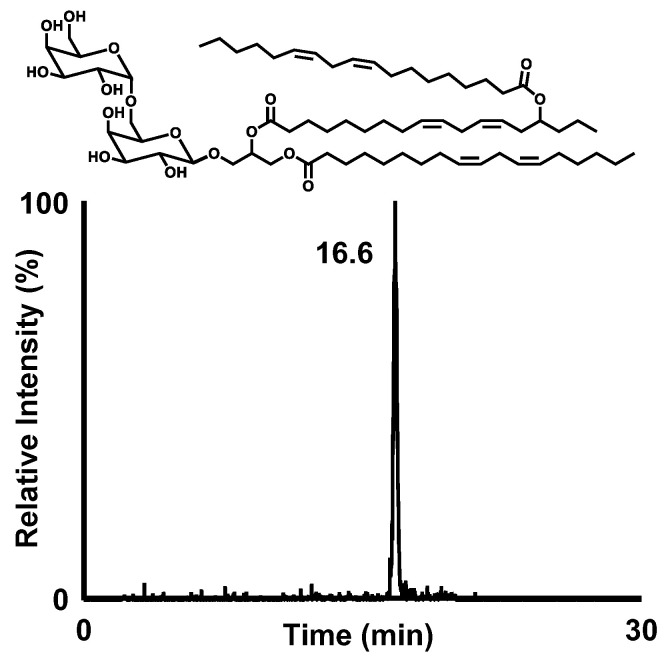
LC-MS/MS analysis of DGDG-LAHLA in the treated cell.

**Table 1 nutrients-14-04153-t001:** Analytical conditions of HPLC-UV, HPLC-MS, and HPLC-MS/MS.

Parameter	Condition 1	Condition 2	Condition 3	Condition 4
Column	Column A	Column A	Column B	Column C
Mobile phase	M.P. A	M.P. B	M.P. C	M.P. D
Flow rate	20 mL/min	20 mL/min	1 mL/min	1 mL/min
Oven temperature	40 °C			
UV	210 nm	210 nm	-	-

Column A, Inertsil ODS-3 10 µm (20 × 250 mm, GL Sciences; Tokyo, Japan); Column B, COSMOSIL Packed Column 5C18-MS-II (2.0 ID × 250 mm, NACALAI TESQUE, INC., Kyoto, Japan); Column C, COSMOSIL Packed Column 5C18-MS-II (2.0 ID × 150 mm, NACALAI TESQUE, INC.); M.P. A, MeOH–2-propanol (9:1, *v*/*v*); M.P. B, MeOH–H2O (97:3, *v*/*v*); M.P. C, MeOH–2-propanol–acetic acid (90:10:0.1, *v*/*v*/*v*); M.P. D, a binary solvent system (A, MeOH–2-propanol–acetic acid (90:10:0.1, *v*/*v*/*v*); B, MeOH–H2O–acetic acid (80:20:0.1, *v*/*v*/*v*) with the following gradient profile (B%): 0–8 min (0%); 8–20 min (100%); 20–30 min (0%)). Condition 1 was used for semi-preparative HPLC-)-UV of glycolipid fraction. Condition 2 was used for semi-preparative HPLC-UV of crude compound X. Condition 3 was used for HPLC-MS analysis of compound X. Condition 4 was used for HPLC-MS/MS of compound X treated with lipase.

**Table 2 nutrients-14-04153-t002:** Analytical conditions of MS and MS/MS.

Parameter	Condition 1	Condition 2
Analysis mode	Q1 mass scan	Product ion scan
Source	ESI	ESI
Polarity	Positive	Positive
Scan range (*m/z*)	50–2000	50–2000
End plate offset (V)	−500	−500
Capillary	4500	4500
Nebulizer (Bar)	1.6	1.6
Dry gas (L/min)	6	6
Dry temp (°C)	180	180
Collision RF (Vpp)	1100	600
Precursor ion (*m/z*)	-	1241.826
Collision energy (V)	-	80

Condition 1 was used for Q1 and LC-MS analysis of compound X. Condition 2 was used for MS/MS analysis of compound X. ESI, electrospray ionization; RF, radio frequency.

**Table 3 nutrients-14-04153-t003:** Analytical conditions of HPLC-MS/MS.

Parameter	Value
Analysis mode	Multiple reaction monitoring (MRM)
Source	ESI
Polarity	Positive
Precursor ion (*m*/*z*)	1241.8
Product ion (*m*/*z*)	961.5
Curtain gas (psi)	20
Ion spray voltage (V)	5500
Turbo gas temperature (oC)	600
Ion source gas 1 (psi)	40
Ion source gas 2 (psi)	80
Collision-activated dissociation gas (psi)	9
Declustering potential (V)	266
Entrance potential (V)	10
Collision energy (V)	81
Collision cell exit potential (V)	24

ESI, electrospray ionization.

## Data Availability

All of the data is contained within the article and the Appendix A.

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
