# Peer review of "Structural Analysis and Anti-Inflammatory Effect of a Digalactosyldiacylglycerol-Monoestolide, a Characteristic Glycolipid in Oats"

_nutrients, 2022, doi:10.3390/nu14194153_

Round 1

Reviewer 1 Report

Structural analysis of GDGD-monoestolide contained in fat-soluble components of oats was carried out, and GDGD-LAHLA was identified as the main one. In addition, by measuring its physiological activity, it showed a synergistic anti-inflammatory effect as a compound substance in LPS-stimulated RAW264 cells and suggested that the effect may be due to the GDGD-LAHLA metabolites incorporated into cells. Academic progress is also evident. On the other hand, to make the results of this research more valuable, please confirm the following points.

1) As a minor point, I think the n-number is 6 in the NO production test and 4 in the anti-inflammatory cytokine production test. I think that the accuracy of the experimental design in this research will increase if there is an explanation as to why the number of n was changed between the two tests. Also, I think it would be good if the number of n in each test was specified in the method.

2) When testing the bioactivity of GDGD-LAHLA, this substance was added during cell culture. Is it correct to understand that this GDGD-LAHLA was purified for structural analysis? In that case, I think that it should be clearly stated at what concentration it was prepared, its degree of purification, and whether it was free from degradation products.

3) For experiments on GDGD-LAHLA uptake into cells, cells were harvested after 24 h co-culture and washed three times with PBS, and GDGD-LAHLA was produced from the cell lysate and analyzed by LC-MS/MS. I have a question about the analysis and presenting the quantity. First, in Fig. 6, the structural formula of DGDG is shown and the retention time is 16.6, which seems to be accompanied by the structure of fatty acids in Fig. 2C. I may be misunderstanding this point, but could you please explain? Was it GDGD-LAHLA or GDGD that was analyzed?

4) In relation to the above, if you are measuring GDGD-LAHLA, if GDGD-LAHLA is adsorbed to the surface of the culture dish, or if it is adsorbed to the surface of the cell, can such GDGD-LAHLA be washed off by washing with PBS? If there is adsorption of either of way and GDGD-LAHLA does not come off after washing with PBS, I think the results will be the same as the findings presented this time. What do you think? Please explain this point as well.

5) A minor question, what percentage of the GDGD-monoestolides in the fat-soluble components of oats is GDGD-LAHLA? I think it would be helpful to understand the value of oats as a functional food if there is a notation about that.

Author Response

Response to Reviewer 1 Comments

Reviewer’s comment: Structural analysis of GDGD-monoestolide contained in fat-soluble components of oats was carried out, and GDGD-LAHLA was identified as the main one. In addition, by measuring its physiological activity, it showed a synergistic anti-inflammatory effect as a compound substance in LPS-stimulated RAW264 cells and suggested that the effect may be due to the GDGD-LAHLA metabolites incorporated into cells. Academic progress is also evident. On the other hand, to make the results of this research more valuable, please confirm the following points.

Response: We wish to express our appreciation to the Reviewer 1 for the positive comments.

Point 1: As a minor point, I think the n-number is 6 in the NO production test and 4 in the anti-inflammatory cytokine production test. I think that the accuracy of the experimental design in this research will increase if there is an explanation as to why the number of n was changed between the two tests. Also, I think it would be good if the number of n in each test was specified in the method.

Response 1: We thank the reviewer for attentions to clarify the details. We used n=4 for the ELISA test because the samples were limited. As suggested by the reviewer, we added information on the number of n in the method section as follows:

    The sentence “RAW264 cells were seeded in 96-well plates at a density of 1.0 × 104 cells/well.” was changed to “RAW264 cells were seeded in 96-well plates at a density of 1.0 × 104 cells/well (n=6).” (Revised manuscript, L163).

    The sentence “Cell viability was determined using the WST-1 assay.” was changed to “Cell viability was determined using the WST-1 assay (n=6).” (Revised manuscript, L174).

    The sentence “RAW264 cells were seeded in 96-well plates at density of 1.0 × 104 cells/well for IL-6 and TNF-α, and 2.0 × 104 cells/well for IL-10.” was changed to “RAW264 cells were seeded in 96-well plates at a density of 1.0 × 104 cells/well for IL-6 and TNF-α, and 2.0 × 104 cells/well for IL-10 (n=4).” (Revised manuscript, L178).

Point 2: When testing the bioactivity of GDGD-LAHLA, this substance was added during cell culture. Is it correct to understand that this GDGD-LAHLA was purified for structural analysis? In that case, I think that it should be clearly stated at what concentration it was prepared, its degree of purification, and whether it was free from degradation products.

Response 2: Thank you for this valuable comment. The cell culture study was performed using the samples which were used for structural analysis. As shown in the result of 1H NMR (Supplementary Information 1), no impurities were identified and the purity reached >99%. The stock solution of the sample was prepared by dissolving it in EtOH to reach a concentration of 3.3 mM.

    With respect to this information, the sentence “DGDG-LAHLA used in this experiment was similar to the one used in Section 2.3. (Structural analysis of compound X).” was added (L165).

    Also, the sentence “These results of MS and NMR analysis also showed that the purity of isolated DGDG-LAHLA reached >99%, which was dissolved in ethanol for use in cell culture study.” was added to the manuscript (Revised manuscript, L296).

Point 3: For experiments on GDGD-LAHLA uptake into cells, cells were harvested after 24 h co-culture and washed three times with PBS, and GDGD-LAHLA was produced from the cell lysate and analyzed by LC-MS/MS. I have a question about the analysis and presenting the quantity. First, in Fig. 6, the structural formula of DGDG is shown and the retention time is 16.6, which seems to be accompanied by the structure of fatty acids in Fig. 2C. I may be misunderstanding this point, but could you please explain? Was it GDGD-LAHLA or GDGD that was analyzed?

Response 3: We incorrectly provided the structural formula for DGDG instead of DGDG-LAHLA in Fig. 6. We replaced it with the correct structural formula for DGDG-LAHLA to correspond with the results of DGDG-LAHLA measurement in Fig. 6. In addition, the retention time in Fig. 2C was also incorrect, and we corrected this in the revised manuscript.

Point 4: In relation to the above, if you are measuring GDGD-LAHLA, if GDGD-LAHLA is adsorbed to the surface of the culture dish, or if it is adsorbed to the surface of the cell, can such GDGD-LAHLA be washed off by washing with PBS? If there is adsorption of either of way and GDGD-LAHLA does not come off after washing with PBS, I think the results will be the same as the findings presented this time. What do you think? Please explain this point as well.

Response 4: Thank you for your comments. It is important to consider whether the DGDG-LAHLA detected in the cell was adhered to the cell membrane or taken up by the cells. Since we did not analyze the membrane and cytoplasm separately, we would like to include the possibility of DGDG-LAHLA adhering to the surface of the petri dish in future studies. We incorporated this information by adding the following sentence: “It is important to consider whether the DGDG-LAHLA detected in the cell was localized in the cytoplasm or membrane-bound. Since we did not separately analyze the DGDG-LAHLA content in membrane and cytoplasm, we would like to specify the cellular localization of the DGDG-LAHLA in future studies.” (Revised manuscript, L409).

Point 5: A minor question, what percentage of the GDGD-monoestolides in the fat-soluble components of oats is GDGD-LAHLA? I think it would be helpful to understand the value of oats as a functional food if there is a notation about that.

Response 5: Thank you for this valuable comment. In this study, the glycolipid fraction (about 9300 mg) was isolated from oats, followed by purification of the DGDG fraction (about 4700 mg), DGDG-LAHLA (about 700 mg), and DGDG-monoestolides other than DGDG-LAHLA (about 800 mg). The results suggested that DGDG-LAHLA made up about half of the DGDG-monoestolides fraction. Still, we could not provide an accurate ratio since not all fractions were recovered from the purification stage, and the amount recovered was reduced to improve the purity. We recognized the importance of this information as there have been no reports on the percentage of DGDG-LAHLA in the DGDG-monoestolides fraction.

    To address this concern, we revised the sentence “Hence, analyses of DGDG-monoestolide molecular species and DGDG-diestolides, leading to the further elucidation of the characteristic lipid profile of oats, are expected in future studies.” to “Hence, analyses of DGDG-monoestolide molecular species and DGDG-diestolides, leading to the further elucidation of the characteristic lipid profile of oats (e.g., the ratio of DGDG-LAHLA in the DGDG-monoestolides fraction of oats), are expected in future studies.” (Revised manuscript, L308).

Reviewer 2 Report

This research type manuscript titled "Structural analysis and anti-inflammatory effect of a digalactosyldiacylglycerol-monoestolide, a characteristic glycolipid in oats”.

In this study the research strategy, methods, results and discussion are clearly and properly supporting the conclusions reached, but it is needed some minor revisions:

1.     WST-1 assay is not a direct measure of cell viability and should be addressed as such. You can be used an assay more specifically for cell viability, and one assay only used for this aim, because if you used one that measure cell proliferation and cell viability it is difficult understand if you exactly measure the viability. 

2.     It is not clear for what the authors intend to use the proposed method. How do the authors prospect this relatively simplistic method of treatment they propose can be improved in the future

3.     In the final conclusion section, limitations and current challenges of this research field should be discussed in a meaningful manner. 

4.     In page 15, line 503, please revise the reference 23, the year was missed. And in my opinion it is important include references more recent (last 5years).

Author Response

Response to Reviewer 2 Comments

Reviewer’s comment: This research type manuscript titled "Structural analysis and anti-inflammatory effect of a digalactosyldiacylglycerol-monoestolide, a characteristic glycolipid in oats”. In this study the research strategy, methods, results and discussion are clearly and properly supporting the conclusions reached, but it is needed some minor revisions:

Response: We wish to express our appreciation to the Reviewer 2 for the positive comments.

Point 1: WST-1 assay is not a direct measure of cell viability and should be addressed as such. You can be used an assay more specifically for cell viability, and one assay only used for this aim, because if you used one that measure cell proliferation and cell viability it is difficult understand if you exactly measure the viability.

Response 1: We thank the reviewer for attentions to clarify the details. The main aim of this experiment is to confirm that the changes in NO production upon treatment with test samples (DGDG-LAHLA) do not correlate with the changes in cell viability. Thus, we indirectly evaluated this using WST-1. In support of this data, our microscopic observation also showed that DGDG-LAHLA does not induce cell death. As Reviewer 2 stated, WST-1 is not a direct method to evaluate cell viability; for this reason, we would like to consider other methods for assessing cell viability in our future studies.

Point 2: It is not clear for what the authors intend to use the proposed method. How do the authors prospect this relatively simplistic method of treatment they propose can be improved in the future.

Response 2: Thank you for your comments. The result from the cell culture experiment in this study showed the promising health benefit (i.e., anti-inflammatory effect) of DGDG-LAHLA. In the future, we believe that it is necessary to corroborate this finding by evaluating the functions of DGDG-LAHLA in vivo, for instance, via human studies.

Point 3: In the final conclusion section, limitations and current challenges of this research field should be discussed in a meaningful manner.

Response 3: Thank you for this valuable comment. After reviewing the past studies and our study, we recognized the remaining limitations in this research area, such as 1) the lack of information on other DGDG-monoestolides besides DGDG-LAHLA despite their high abundance in oats; 2) the unknown ratio of DGDG-LAHLA to DGDG-monoestolides in oats, and 3) where DGDG-LAHLA and its degradation products are localized in cells (membrane, cytosol, etc.). Future studies should focus on these areas to gain a more comprehensive understanding of the role of oat-derived DGDG-monoestolides, including DGDG-LAHLA, in our health.

    With respect to this information, we revised the sentence “Hence, analyses of DGDG-monoestolide molecular species and DGDG-diestolides, leading to the further elucidation of the characteristic lipid profile of oats, are expected in future studies.” to “Hence, analyses of DGDG-monoestolide molecular species and DGDG-diestolides, leading to the further elucidation of the characteristic lipid profile of oats (e.g., the ratio of DGDG-LAHLA in the DGDG-monoestolides fraction of oats), are expected in future studies.” (Revised manuscript, L308). And we added the sentence “It is important to consider whether the DGDG-LAHLA detected in the cell was localized in the cytoplasm or membrane-bound. Since we did not separately analyze the DGDG-LAHLA content in membrane and cytoplasm, we would like to specify the cellular localization of the DGDG-LAHLA in future studies.” (Revised manuscript, L409).

Point 4: In page 15, line 503, please revise the reference 23, the year was missed. And in my opinion it is important include references more recent (last 5years).

Response 4: We revised reference 23 and added the publication year as suggested (Schröter, D.; Neugart, S.; Schreiner, M.; Grune, T.; Rohn, S.; Ott, C. Amaranth’s 2-Caffeoylisocitric Acid-An Anti-Inflammatory Caffeic Acid Derivative That Impairs NF-ΚB Signaling in LPS-Challenged RAW 264.7 Macrophages. Nutrients, 2019, 11(3). 571. https://doi.org/10.3390/nu11030571.). (Revised manuscript, P?L?). This particular study was published in 2019.

Round 2

Reviewer 1 Report

The author responds appropriately to comments. For this reason, I think that the academic value of the revised version has increased.